# Using the Finite Element Method to Determine the Odonto-Periodontal Stress for a Patient with Angle Class II Division 1 Malocclusion

**DOI:** 10.3390/diagnostics13091567

**Published:** 2023-04-27

**Authors:** Mahmoud Katta, Stelian-Mihai-Sever Petrescu, Lucian Paul Dragomir, Mihai Raul Popescu, Ruxandra Voinea Georgescu, Mihaela Jana Țuculină, Dragoș Laurențiu Popa, Alina Duță, Oana Andreea Diaconu, Ionela Teodora Dascălu

**Affiliations:** 1Department of Orthodontics, Faculty of Dental Medicine, University of Medicine and Pharmacy of Craiova, 200349 Craiova, Romania; katta_dent@icloud.com (M.K.); marceldascalu@yahoo.com (I.T.D.); 2Department of Occlusology and Fixed Prosthetics, Faculty of Dental Medicine, University of Medicine and Pharmacy of Craiova, 200349 Craiova, Romania; dragomirlucianpaul@yahoo.com (L.P.D.); popescumihairaul@yahoo.com (M.R.P.); 3Department of Periodontology, Faculty of Dental Medicine, “Titu Maiorescu” University of Bucharest, 031593 Bucharest, Romania; ruxi0372@yahoo.com; 4Department of Endodontics, Faculty of Dental Medicine, University of Medicine and Pharmacy of Craiova, 200349 Craiova, Romania; mtuculina@yahoo.com (M.J.Ț.); oanamihailescu76@yahoo.com (O.A.D.); 5Department of Automotive, Transportation and Industrial Engineering, Faculty of Mechanics, University of Craiova, 200478 Craiova, Romania; alina.duta@edu.ucv.ro

**Keywords:** malocclusion, finite element method, orthodontic system, periodontal ligament, result maps

## Abstract

The finite element method (FEM) is a computational method that can solve all biomechanical problems, including the field of orthodontics. The purpose of this virtual experimental study is to determine the behavior of a real orthodontic system subjected to different systems of loads. To analyze the real orthodontic system, we studied the case of a 21-year-old female patient. We used the InVesalius program, which can transform a set of DICOM-type images taken from cone beam computed tomography (CBCT) into three-dimensional structures. These structures were edited, modified, completed, and analyzed from a geometric point of view with the help of the Geomagic software. The final result of these operations must be a three-dimensional model made up of perfectly closed surfaces so that they can be transformed into virtual solids. The model consisting of perfectly closed surfaces is loaded into computer-aided design (CAD) programs. Bracket and tube components, as well as orthodontic wires, can be added to these models, similar to the analyzed patient’s tissues. When the model is complete and geometrically correct, it is exported to a program that uses FEM, such as Ansys Workbench. The simulation was performed for the forces of 0.5, 0.6, 0.7, 0.8, 0.9, and 1 N. The intention was to determine the behavior of the entire orthodontic system for these force values. After running the simulations, result maps were obtained that were composed of displacement, strain, and stress diagrams. It was also found that, in addition to the known rigidity, the orthodontic system has some elasticity due to the orthodontic wires, as well as the periodontal ligaments. Thus, a virtual analysis study can be carried out starting from a real patient with pre-treatment CBCT images and the virtual models of the bracket and tube elements and of the orthodontic wires.

## 1. Introduction

The etiopathogenesis of malocclusions is complex and multifactorial, including general, phylogenetic, hereditary, endocrine, dysmetabolic, and loco-regional factors [1].

The ever-increasing prevalence of malocclusions led the World Health Organization (WHO) to consider this pathology as the third most common oral health problem after caries and periodontal diseases [2]. In addition, the WHO refers to malocclusions as a disability due to the impairment of the normal functions of the dentomaxillary apparatus, particularly physiognomy. In this way, the stress to which the individual and his family are subjected was solved. In modern men, the correction of malocclusions has become a necessity in his social behavior. Studies in the specialized literature have shown that a patient esthetically and functionally rehabilitated through orthodontic treatment became much more socially attractive than one with malocclusion [3]. A successful orthodontic therapy aims to achieve a correlation between facial aesthetics and normal occlusion. Thus, it is necessary to implement more innovative methods of diagnosis and treatment.

CBCT is a three-dimensional imaging investigation method that has revolutionized diagnosis and treatment planning in orthodontics. Thus, after performing CBCT, an increased bone density was observed at the level of the mid-palatal suture in a group of patients who presented Angle Class II malocclusions. It was also demonstrated that orthodontic diagnosis can be changed [4,5,6]. Another study performed on two groups of 60 cases selected from 137 CBCT scans obtained from patients with and without an open bite (each group including adults of both genders and approximately equal mean age) demonstrated how two different three-dimensional software packages (Planmeca Romexis and Nemotec NemoStudio) used in CBCT interpretation affect the volumetric and cross-sectional measurements of the oropharyngeal airway, particularly in individuals without an open bite [7]. In addition, recent research indicates the correlation between skeletal Class II malocclusion and the position of the mandibular condyle affecting the temporomandibular joint (TMJ) and breathing. The authors of this study emphasize the importance of starting treatment of TMJ disorders as soon as this link is noted [8].

Initially, the finite element method (FEM) was used to solve the most complex problems in the field of continuous elastic structures, from civil, industrial, or dam constructions to the construction of ships and spacecraft and, recently, the medical field for the simulation of biomechanical systems. Thus, this method also has applicability in orthodontics [9,10,11,12].

In the finite element method, as a starting point, an integral model of the studied phenomenon is used. It is applied separately for a series of small regions of a continuous structure obtained by the discretization process, called finite elements, connected to each other by points, called nodes. These finite elements must be designed in such a way that their ensemble reconstructs as faithfully as possible the real structure analyzed. In principle, these links must be designed in such a way as to allow numerical convergence to the exact solution, when the structure is discretized in finite elements with increasingly smaller dimensions [13].

The goal of orthodontic treatment is not only to improve the functionality of the dentomaxillary apparatus and facial aesthetics but also to maintain the health of the periodontal structures. Regardless of the therapeutic performance of an orthodontist, a correction of a malocclusion can represent a failure if periodontal susceptibility is not considered. The successful results of orthodontic treatment, both short-term and long-term, are influenced by periodontal status before, during, and after fixed orthodontic therapy, which also includes contention after active therapy.

The purpose of the study is to determine the behavior of a real orthodontic system subjected to different load systems. It should be specified that, in the analyzed model, the bony components of the mandible and maxilla, the periodontal ligaments, teeth, orthodontic wires, and the bracket and tube components on which forces less than 1 N act are included [14].

Studying the specialized literature, no significant research was found to analyze the mechanical behavior of an orthodontic system. However, even if a FEM study is done only on one tooth, it is highlighted that the forces in the orthodontic system have a value of 1 N or less [11]. In addition, similar conclusions resulted from another study, the force values that appear being approximately 1 N [12]. At the same time, in a recent study, a system consisting of a simplified mandible and four teeth was analyzed, testing forces between 0.25 and 5 N [14]. These three studies were carried out on simple three-dimensional models, so not on systems similar to those of the patients, having mainly theoretical significance. A more complex study, carried out on a model similar to the model of a patient, determined the forces that appear in the entire orthodontic system, having a value between 0 and 5 N [15].

In addition, we were interested in checking if there is a functional correlation between the value of forces acting on the bracket and tube elements and the maximum values of displacement, strain, and stress and if the application of forces through the orthodontic wire can cause damage to the periodontal ligaments.

## 2. Materials and Methods

The present study was approved by the Ethics Committee of the University of Medicine and Pharmacy of Craiova, Romania (approval reference no. 55/11 April 2022), in accordance with the ethical guidelines for research with human participants of the University of Medicine and Pharmacy of Craiova, Romania. Written informed consent was obtained from the subject involved in the study.

A part of the data used in this study was obtained from a research project (no. 26/527/27 May 2022) carried out in partnership with the University of Medicine and Pharmacy of Craiova.

To carry out this virtual experimental study, we studied the case of a 21-year-old female patient who presented to the Orthodontic Clinic of the Faculty of Dental Medicine at the University of Medicine and Pharmacy of Craiova.

Extraoral and intraoral photos of the subject were taken. The photos were made with a DSLR 600EOS camera (Canon, Ota, Tokyo, Japan). Figure 1 shows images of the subject.

The patient underwent bimaxillary CBCT. To obtain tomographic images, we used a CS 8200 3D CT scanner (Carestream Dental, Atlanta, GA, SUA). Figure 2 shows a CBCT image of the maxilla and mandible, obtained with OnDemand3D Dental (Cybermed Corp., Yusong, Republic of Korea), a software tool used in dental imaging.

The inclusion criteria in our study were based on the complexity of the clinical picture of the selected patient, who was diagnosed with Angle Class II Division 1 malocclusion and maxillary compression with protrusion. The tooth movements that occur during orthodontic treatment with fixed appliances are essential in determining the behavior of the entire orthodontic system.

Before the fixed orthodontic treatment, the odontectomies of 3.8 and 4.8 were performed in order to prevent post-treatment relapse. The surgical interventions were performed in the Oral and Maxillofacial Surgery Clinic of the Emergency County Clinical Hospital of Craiova. Photos from the time of 3.8 odontectomy were also taken. Figure 3 shows images of the surgical intervention.

Fourteen days after the odontectomies, the sutures were suppressed. The following day, the fixed orthodontic appliance was bonded using the straight-wire technique. We chose this method because it reduces the treatment time by moving the dental assembly in all three spatial planes. The straight-wire technique uses pre-angled brackets and modern memory wires, which enable sliding tooth movements, thus ensuring safe guidance.

Initially, a medical imaging program, OnDemand3D Dental, was used for visual analysis of the patient’s dentomaxillary apparatus, which, in addition to the specific instruments, offered three-dimensional images. However, this software did not allow the generation of a virtual model that could later be analyzed with different tools, such as the finite element method.

For this reason, the InVesalius software (CTI, Campinas, Brazil) was used, which could transform a set of DICOM images into three-dimensional structures known in engineering as “point clouds”. InVesalius is an open-source program dedicated to medical research that, based on shades of gray, offers three-dimensional geometries for different tissues (fat tissue, muscle, bone, enamel, etc.).

These “point cloud” structures were transferred to the Geomagic software (Morrisville, NC, USA), which allows the three-dimensional geometry to be transformed into perfectly closed surfaces. Geomagic is a piece of software dedicated to reverse engineering that allows the editing of “point cloud” geometric structures. It can also remove “artifact-type” surfaces or non-conforming components. The final goal is to transform these closed surfaces into “virtual solid-type” geometries that can be analyzed with programs that use the finite element method, and the biomechanical behavior of the analyzed structures can be determined.

In Geomagic, “point cloud” structures can also contain elements that do not belong to the real patient, called “artifacts”. These are due to the phenomena of refraction and reflection of X-rays when they pass through different structures. These artifacts can be removed in this program. The “point cloud” is first transformed into initial spatial triangular surface structures by “coupling” three adjacent points that make up the point cloud. A regular CBCT can generate a model that may contain several million such initial triangular surfaces. In Geomagic, structures can be edited, modified, completed, and analyzed from a geometrical point of view. In a regular model, which comes from a patient, there can be many gaps that, in reality, do not exist. These gaps can be “filled” virtually without affecting the real geometry. Non-conforming surfaces can also be determined and transformed into conforming ones. The final result of operations based on the reverse engineering method must be a three-dimensional model consisting of perfectly closed surfaces so that they can be transformed into virtual solids. In reality, from a geometric point of view, it is known that human tissues are perfectly closed surfaces.

Next, the model consisting of perfectly closed surfaces is loaded into CAD programs where, initially, the surfaces are transformed into models made up of virtual solids. To these models, considered similar to the analyzed patient’s tissues, components specific to orthodontic therapy can be added (in our case, bracket and tube components, as well as orthodontic wires). These components are defined in the parameterized virtual environment using CAD methods and are attached to the virtual model of the studied patient. When the model is complete and geometrically correct, meaning the virtual solids do not interfere, it is exported to a program that uses the finite element method.

One such program is Ansys Workbench (Ansys Inc., Canonsburg, PA, USA), in which a system subjected to various physical phenomena can be analyzed, such as mechanical loading, elasticity of materials, and thermal radiation. In the studied case, we were interested in the mechanical phenomena that appear in the components of an orthodontic system made up of virtual elements specific to the patient, but also in the specific components of orthodontic treatment with fixed appliances, such as orthodontic wires and bracket and tube elements. In this program, the material properties of both the biological components (bones, periodontal ligaments, and dental enamel) and the added components (brackets, tubes, and orthodontic wires) were assigned to the virtual components. The areas considered fixed were indicated, namely portions of the jaw and mandible, and their forces and position were established. Finally, after running the simulations, result maps composed of strain and stress diagrams were obtained.

### 2.1. The Virtual Model of the Dentomaxillary Apparatus of the Studied Patient Obtained from CBCT Images

The primary three-dimensional reconstruction was carried out using the open-source program InVesalius, which is dedicated to medical research. For this, two main filters were used, namely bone and enamel, to obtain the geometries for the bone components and the structure of the teeth. Initially, in InVesalius, a structure made up of “point clouds” was obtained, similar to the three-dimensional scan specific to reverse engineering. At first, the intention was to obtain the two main bone components, namely the fragments of the maxilla and the mandible. These geometries based on point clouds were loaded into Geomagic, as shown in Figure 4. Using an algorithm based on reverse engineering methods, the structure based on “point clouds” was transformed into 4,903,473 elementary triangular surfaces.

It can be observed that, in addition to “artifact-type” elements, the initial model contained the bone components of the cervical spine and dental structures that appeared in the model due to the shades of gray similar to the bone structures. These structures and elements were removed using different procedures. Unlike the models obtained from three-dimensional scanning, where the structures were bordered by a single outer surface, in the models obtained from CBCT images, surfaces were found in several layers, so the removal operations were quite laborious. Mainly, the lasso-type selection and the direct removal method were chosen. Figure 5 shows some stages of the removal operations.

A series of “artifacts” were identified in this primary model, which obviously did not belong to the real patient. In this case, they were found in the area of the teeth, and these elements were removed using the same techniques. The stages of this operation are presented in Figure 6. At this stage, the model had 2,109,072 elementary triangular surfaces.

The aim was to obtain the bone components of the analyzed patient separately in the first phase. Due to the close density of the teeth compared to the bone components, they were recovered in the model, even if a Bone-type filter was used in InVesalius. For this reason, these three-dimensional structures that made up the model at that time were also removed. Figure 7 shows the elimination stages of these structures.

To complete the model, operations were applied to “fill the gaps” to reduce the number of triangular surfaces coupled with successive “finishing” operations and, finally, to obtain a correct model without non-conforming surfaces using different techniques from reverse engineering. In the end, the model had 87,962 elementary triangular surfaces and was composed of perfectly closed surfaces without non-conforming elements and without self-intersections. Figure 8 shows the model of the two bone components subjected to some of these final operations.

In order to obtain the virtual structure of the teeth, the “point cloud” model was used, obtained in InVesalius by using the Enamel-type filter. Figure 9 shows the model of this geometric structure after it was imported into Geomagic for processing. Initially, the model had 1,390,274 elementary triangular surfaces.

As can be seen, the model also showed surfaces that belong to the bone components due to the close density and, implicitly, the close shades of gray. For this reason, the model was subjected to complicated and laborious elimination operations. Figure 10 shows some stages of these operations performed in Geomagic.

Many series not only of elimination of “artifact-type” elements, but also of the presence of bone components were applied, and the resulting model is presented in Figure 11. At that moment, the model had 357,406 elementary triangular surfaces.

An important problem was that of “separating” the virtual teeth so that they work independently in the orthodontic system. For this, successive stages of elimination and “filling” were used. Figure 12 shows some of the stages of these operations.

After several stages, in which numerous reverse engineering techniques were used, it was considered simpler to work on two files that contained the tooth models so that, in the end, these models would “meet again” in a final file. Thus, Figure 13 shows the initial model and the two structures that contained the entire dentition.

At that moment, after numerous finishing operations, the dentition models were obtained. From these models, the primary models of the periodontal ligaments were obtained through an Offset operation at 0.2 mm. These models were later cut out at the level of the bone components. Next, the models of the two bone components were loaded into SolidWorks (SolidWorks Corp., Waltham, MA, USA), and the model was immediately transformed into virtual solids, as shown in Figure 14.

In the Assembly module, the components later used in the simulations with the finite element method were loaded one by one. First, the bone components were loaded, then the two models that represent the initial models (Offset) of the ligaments, as shown in Figure 15.

Using CAD techniques for aligning the planes that compose the coordinate systems, these systems were positioned by applying constraint commands. Next, a similar alignment operation was applied to the tooth models. At that moment, an operation was necessary to cut the initial models of the ligaments at the level of the bone components, as is the case in reality. The internal cavity of the ligaments was created by the virtual “decrease” in the outer surfaces of the teeth. Finally, the obtained model contained the bone components, ligaments, and teeth, as shown in Figure 16.

Petrescu S.M.S. et al. defined the models of orthodontic wires, brackets, and tube elements in the parameterized SolidWorks CAD system and detailed these stages [15]. These models were defined, some in the Assembly context and others separately in the Part module. Finally, all these components were grouped in the Assembly module using specific CAD techniques. Figure 17 shows this virtually identical orthodontic model to the one used in the patient’s first treatment.

### 2.2. Simulation of the Mechanical Behavior of the Orthodontic System with Loads of 0.5–1 N for Each Element of the Bracket Type

The model of the orthodontic system was loaded into the finite element analysis program in Ansys Workbench, and the model is presented in Figure 18.

In a first step, the model was divided into finite elements using default settings. Ansys Workbench realizes the structure of finite elements based on a convergence algorithm. A total of 1,155,872 finite elements with 2,129,656 nodes were obtained. The structure of elements composed of tetrahedrons was obtained, as can be seen in Figure 19.

In order to create a simulation as close to reality as possible, a series of papers were analyzed to determine the physical and mechanical properties of the periodontal ligament [16,17,18,19].

The other materials that compose the orthodontic system are known from another study that Petrescu S.M.S. et al. developed [15]. Table 1 shows the properties of the materials used in the simulations of our study.

In the Engineering Data module, these materials were added to a separate library so that they can be used currently in Ansys Workbench.

In Ansys Workbench, for Static Structural simulations, the system of constraints, called boundary conditions, was defined for the convergence of the solution to results similar to the real case analyzed. The bonded contact surfaces were created automatically. The fixation surfaces found on the upper part of the maxilla and on the lower part of the mandible were defined (in blue). Furthermore, the system of forces that are placed perpendicular to the bracket, and tube elements were defined with the value of 0.5 N for the first simulation. The fixation surfaces and the system of these forces and their location are shown in Figure 20. In total, 24 forces were used, one for each bracket and tube element.

Finally, the simulation was performed for the forces of 0.5 N, and then, it was repeated for the forces of 0.6, 0.7, 0.8, 0.9, and 1 N. On the whole, six such simulations were carried out. The intention was to determine the behavior of the entire orthodontic system for these force values.

## 3. Results

For the loads given by the forces of 0.5 N, a first set of results was obtained that consisted of displacement, strain, and stress maps. In Figure 21, these results are shown.

The results maps obtained on the periodontal ligaments of the system loaded with forces of 0.5 N were also interesting. Figure 22 shows these results for displacement, strain, and stress.

Thus, for the force value of 0.5 N, the following maximum values were obtained:-Maximum displacement was 1.1012 × 10^−5^ m;-Maximum strain was 1.4843 × 10^−3^ m/m;-Maximum stress was 4.894 × 10^7^ Pa.

After resuming the simulation for the value of 0.6 N of the forces placed on each bracket and tube element, the result maps presented in Figure 23 were obtained.

Thus, for the force value of 0.6 N, the following maximum values were obtained:-Maximum displacement was 1.3215 × 10^−5^ m;-Maximum strain was 1.7811 × 10^−3^ m/m;-Maximum stress was 5.8728 × 10^7^ Pa.

Next, the 24 forces were changed to the value of 0.7 N, the application was run again, and the result maps presented in Figure 24 were obtained.

In this case, for the force value of 0.7 N, the following maximum values were obtained:-Maximum displacement was 1.5417 × 10^−5^ m;-Maximum strain was 2.078 × 10^−3^ m/m;-Maximum stress was 6.8516 × 10^7^ Pa.

In order to obtain the next simulation, the values of the forces acting on the bracket and tube elements were changed to the value of 0.8 N, and the result maps from Figure 25 were obtained.

In this simulation, for the force value of 0.8 N, the following maximum values were obtained:-Maximum displacement was 1.762 × 10^−5^ m;-Maximum strain was 2.3748 × 10^−3^ m/m;-Maximum stress was 7.8303 × 10^7^ Pa.

Using the same model from Ansys Workbench but changing the force values to a value of 0.9 N, the result maps presented in Figure 26 were obtained.

After running the application for the force value of 0.9 N, the following maximum values were obtained:-Maximum displacement was 1.9822 × 10^−5^ m;-Maximum strain was 2.6717 × 10^−3^ m/m;-Maximum stress was 8.8091 × 10^7^ Pa.

Finally, the forces acting on the bracket and tube components were changed to a value of 1 N, and after running the simulation, the result maps presented in Figure 27 were obtained.

At the end, for the force value of 1 N applied on the bracket and tube elements, the following maximum values were obtained:-Maximum displacement was 2.2025 × 10^−5^ m;-Maximum strain was 2.9685 × 10^−3^ m/m;-Maximum stress was 9.7879 × 10^7^ Pa.

The maximum values of displacement, strain, and stress were stored in an Excel file for each force value that evolves from 0.5 to 1 N. Thus, Figure 28 shows the displacement, strain, and stress diagrams for the force values that act on the bracket and tube elements.

Diagrams can be obtained based on maximum displacement values. Thus, Figure 29 shows the strain vs. displacement diagram, and Figure 30 shows the stress vs. displacement diagram.

## 4. Discussion

In the diagrams shown in Figure 28, Figure 29 and Figure 30, we illustrated the approximately linear correlation between displacement, strain, and stress. We also showed the value of the applied forces. At the same time, a functional link was determined between displacement, strain, and stress, which shows that the forces that appear in an orthodontic system produce effects that cannot lead to damage of the periodontal ligaments.

In vivo, CBCT analysis is a non-invasive and clinically effective tool used for teeth examination that may ultimately improve the outcome of dental treatments [20].

The CBCT evaluation of the periodontal tissues plays an important role in the diagnosis and development of the orthodontic treatment plan. Under the action of orthodontic forces, gingival recessions may occur in the case of a thin periodontal phenotype. However, CBCT investigations are not part of the usual complementary examinations used in orthodontic therapy [21,22].

There are also mathematical methods for establishing bone levels, such as fractal dimension analysis (FDA). The geometry of bones is very complex. Its shape is complicated to measure using only traditional measuring methods. Traditional measuring techniques on a bone specimen require supplementary measuring. For the FDA, mathematic formulas are used to describe complicated shapes. The FDA offers a possibility of a comparison between complex shapes, such as a histological image of a bone defect [23].

FEM is a non-invasive technique with a high degree of fidelity and is also used in orthodontics. It has applicability in biomechanics, especially in the analysis of odonto-periodontal stress [24].

Indeed, the quality of an FEM-based simulation is influenced by the number of finite elements. We consider that the finite element structure used (1,155,872 finite elements with 2,129,656 nodes) in these simulations is sufficient to obtain good results.

FEM can be used in the modeling and simulation of an orthodontic system with the aim of establishing the elastic forces arising during orthodontic therapy with a fixed appliance and the mechanical effect generated by them. To achieve the objectives, the researchers used CBCT images obtained from a patient with malocclusion, in which fixed orthodontic treatment was subsequently initiated. The three-dimensional processing of metal orthodontic elements included an upper orthodontic wire, a lower orthodontic wire, and a set of brackets and tube components. Through the Ansys Workbench program, a mathematical model of the forces appearing during the orthodontic treatment, following the deformation of a wire, was obtained. Thus, the values of maximum mechanical displacement, maximum deformation, maximum stress, and deformation energy were obtained [15].

Other research has been carried out using FEM, which analyzed the distribution of orthodontic forces between the periodontal ligament and the alveolar bone. Thus, the researchers had the opportunity to reveal the areas where root resorption can occur. The tendency of the teeth to move contrary to the application of orthodontic forces has also been evaluated by means of the Ansys 14 program [25].

A FEM study evaluated the maximum stress, direction of application of forces, and displacement produced by the tooth-periodontal ligament–alveolar bone complex with different degrees of periodontal tissue damage. The applied forces had intensity from a maximum of 1 N. The forces that are generated during an orthodontic treatment with a fixed appliance are difficult to quantify from a clinical point of view. The authors of this study recommend that the intensity of these forces should not exceed 1 N in the case of affected teeth [14].

A finite element analysis determined the effect of different tooth penetration depths into the maxillary sinus floor on the orthodontic force system for bodily tooth movement. The authors claimed that the individual differences in the periodontal structure should be considered during the orthodontic treatment [26].

Another study using FEM analyzed the stress distribution in the periodontal ligament and tooth structure of a cementum-reinforced tooth, a dentine-reinforced tooth, and an immature tooth during orthodontic treatment. The authors used a finite element model of a maxillary incisor and its supporting tissues. After performing this study, they concluded that orthodontic loads can be applied in teeth previously treated with regenerative endodontics [27].

The aim of an investigation using a bracketless orthodontic treatment (BOT) was to calculate the stress magnitude in the central incisor movement through a finite element analysis. The authors used different diameters of nickel–titanium wires and two different resin composites. The results of this study showed that the BOT technique promoted a suitable biomechanical response during central incisor movement [28].

Although there are few studies that use FEM to study the dentomaxillary apparatus based on orthodontic wires, the values of the forces are similar to the forces used in our study [11,12,14,15]. In addition, the effects of their application is almost the same, rendered by displacement, strain, and stress maps. However, our study is based on a much more complicated system, similar to the system of a real patient where the geometry of the virtual tissues is quasi-identical, and this is obtained from the processing of real CBCT images. In addition, in this system, the periodontal ligament was also included in order to obtain a three-dimensional model as close to reality as possible.

The limitations of our study are represented by the fact that not all the soft tissues were taken into account (such as dental pulp and gingival tissue) because, for now, we considered that their effect is not significant.

## 5. Conclusions

Analyzing the results obtained through the six FEM simulations for a real orthodontic system, it was found that, in addition to the known rigidity, the orthodontic system has some elasticity due to the orthodontic wires and to the periodontal ligaments.

Through this study, we demonstrated that functional correlations can be determined between the values of the forces applied on the bracket and tube elements and the maximum values of displacement, strain, and stress.

The maximum values from the stress maps, which indicate the areas of possible damage, are relatively small, which proves once again that the treatment based on the straight-wire technique cannot lead to damage to the dental and periodontal tissues if it is applied according to the known orthodontic protocols.

From the analyses carried on the stress diagrams, it can also be seen that the maximum values are found on the buccal surfaces of the teeth, just below the bracket and tube elements. This observation indicates that enamel cracks may appear in these areas if the fixation of the bracket and tube elements is not done correctly or if the orthodontic system based on the straight-wire technique is maintained for a longer period than is established in the treatment plan.

The present paper demonstrates that a virtual analysis study can be carried out starting from a real patient that includes pre-treatment CBCT images and the virtual models of the bracket and tube elements and of the orthodontic wires.

## Figures and Tables

**Figure 1 diagnostics-13-01567-f001:**
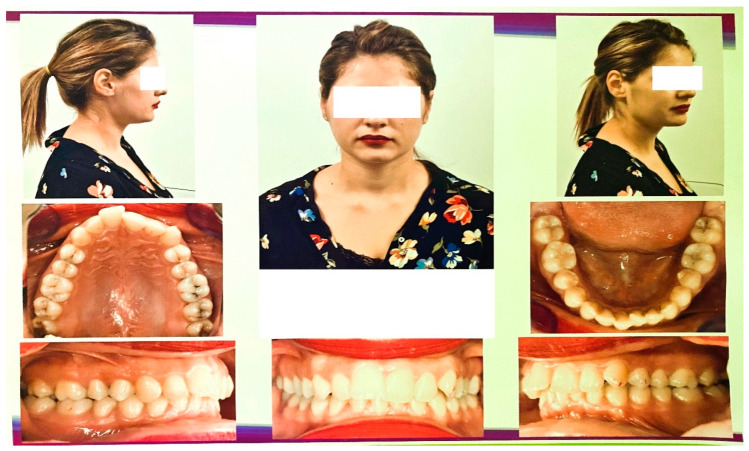
Extraoral and intraoral photos of the subject.

**Figure 2 diagnostics-13-01567-f002:**
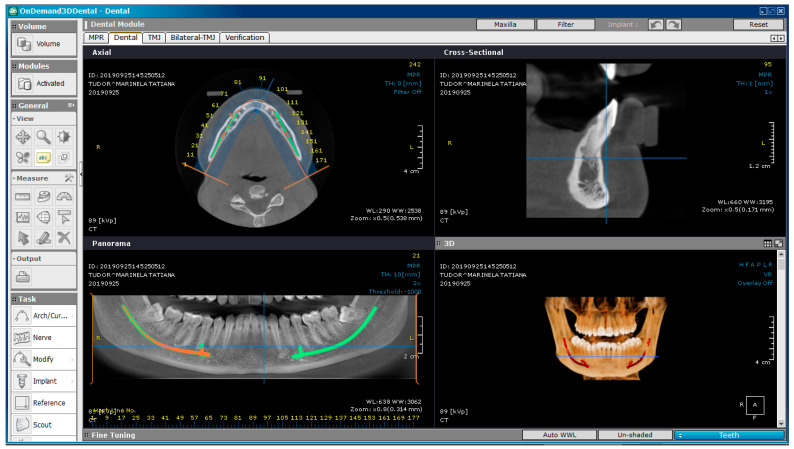
A CBCT image of the subject.

**Figure 3 diagnostics-13-01567-f003:**
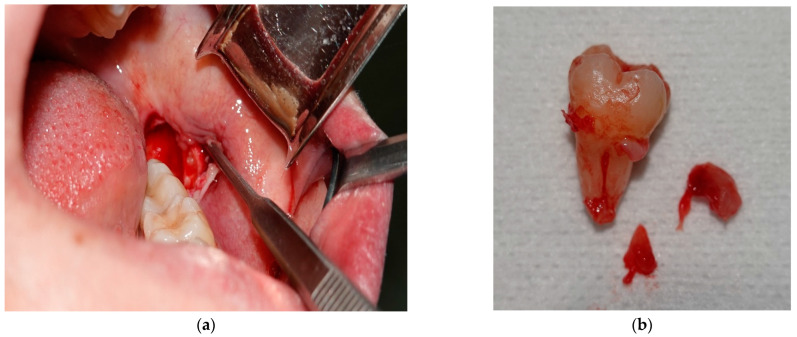
Photos from the surgical intervention (**a**,**b**).

**Figure 4 diagnostics-13-01567-f004:**
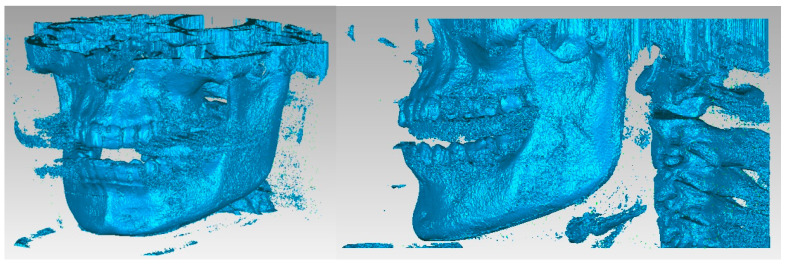
The primary model of the bone components loaded in Geomagic.

**Figure 5 diagnostics-13-01567-f005:**
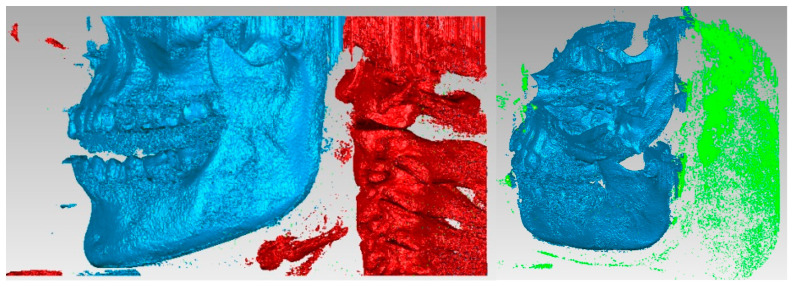
Stages of removing vertebrae in Geomagic.

**Figure 6 diagnostics-13-01567-f006:**
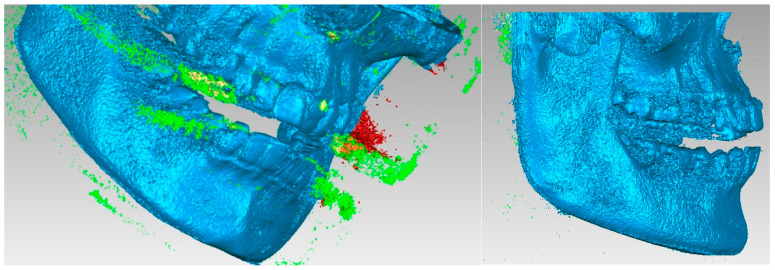
Elimination of some “artifact-type” elements.

**Figure 7 diagnostics-13-01567-f007:**
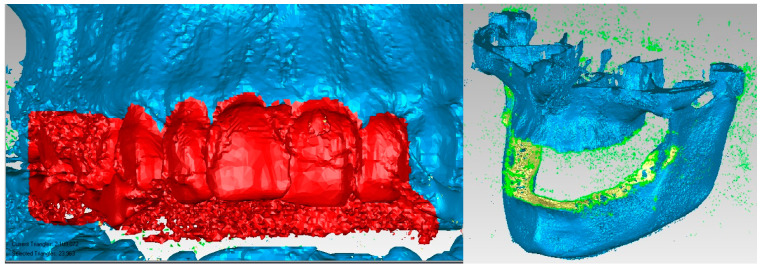
Removing virtual teeth from the bone component model.

**Figure 8 diagnostics-13-01567-f008:**
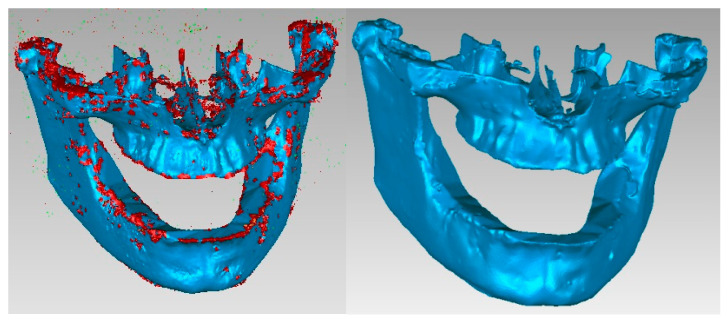
Final stages of reverse engineering to obtain the two virtual bone components.

**Figure 9 diagnostics-13-01567-f009:**
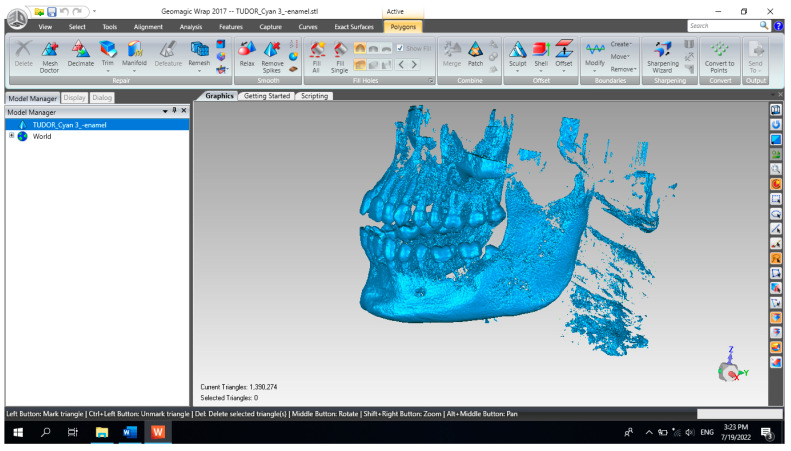
The initial model of the Enamel-type structure in Geomagic.

**Figure 10 diagnostics-13-01567-f010:**
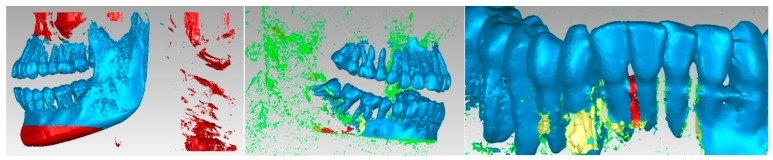
Steps to remove additional structures.

**Figure 11 diagnostics-13-01567-f011:**
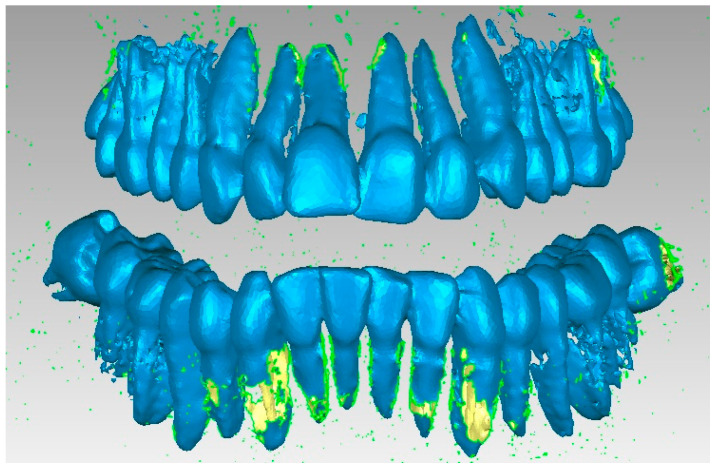
Intermediate model of the dentition of the studied patient.

**Figure 12 diagnostics-13-01567-f012:**
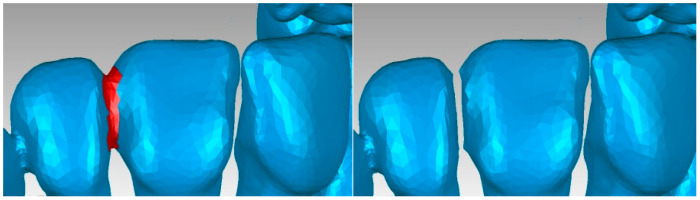
Steps for separating the teeth.

**Figure 13 diagnostics-13-01567-f013:**
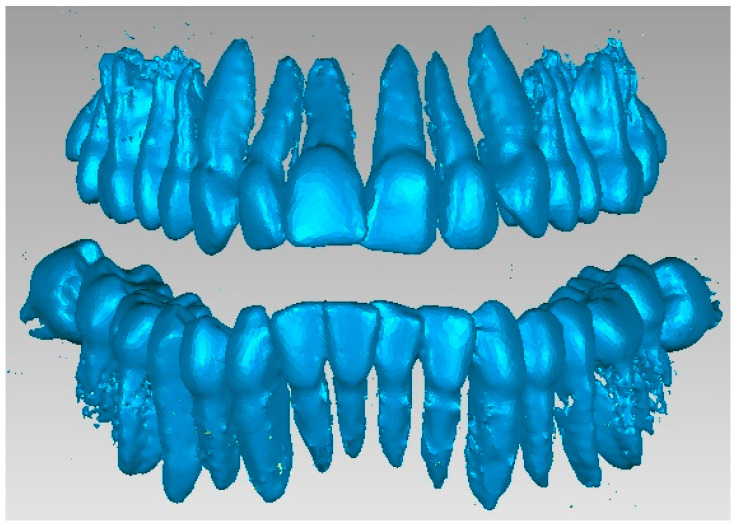
Dentition separation in two virtual models.

**Figure 14 diagnostics-13-01567-f014:**
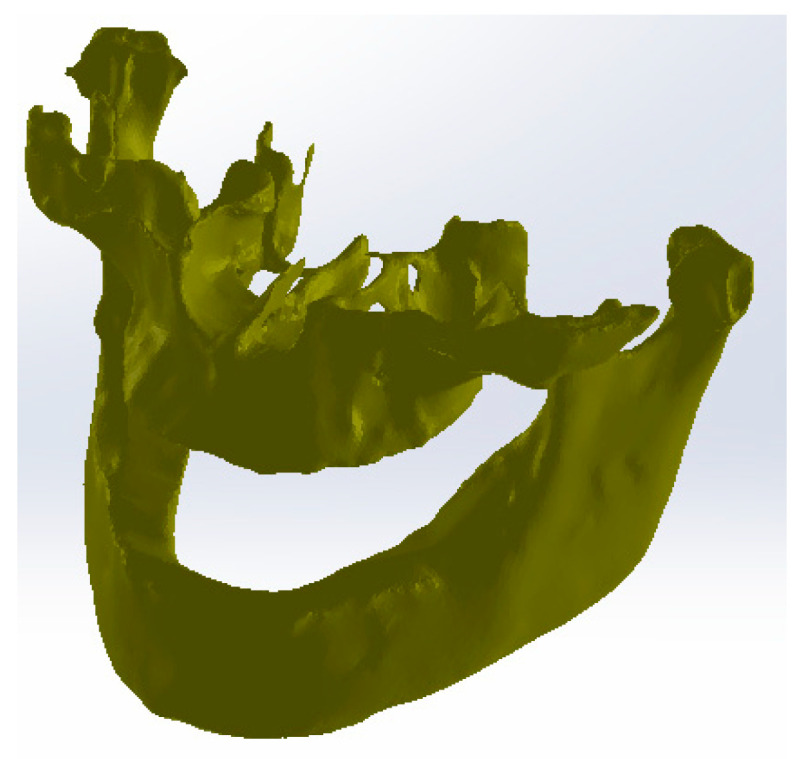
Bone component models in SolidWorks.

**Figure 15 diagnostics-13-01567-f015:**
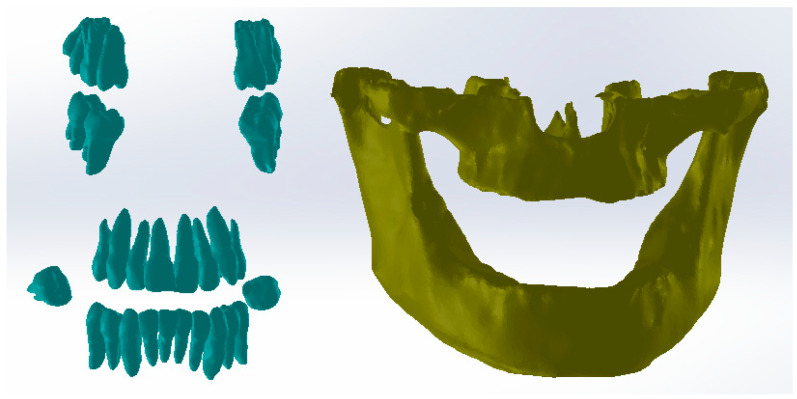
Models of ligaments (blue) and bone components (yellow) in SolidWorks.

**Figure 16 diagnostics-13-01567-f016:**
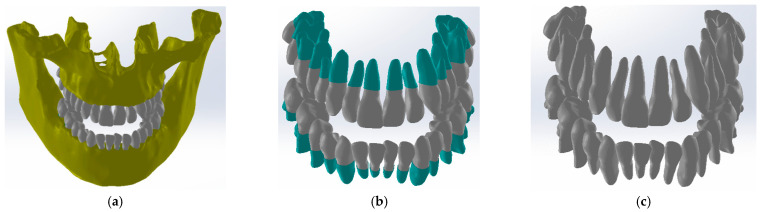
The identical virtual model of the analyzed patient: bone components, periodontal ligaments, and teeth (**a**); periodontal ligaments and teeth (**b**); and teeth (**c**).

**Figure 17 diagnostics-13-01567-f017:**
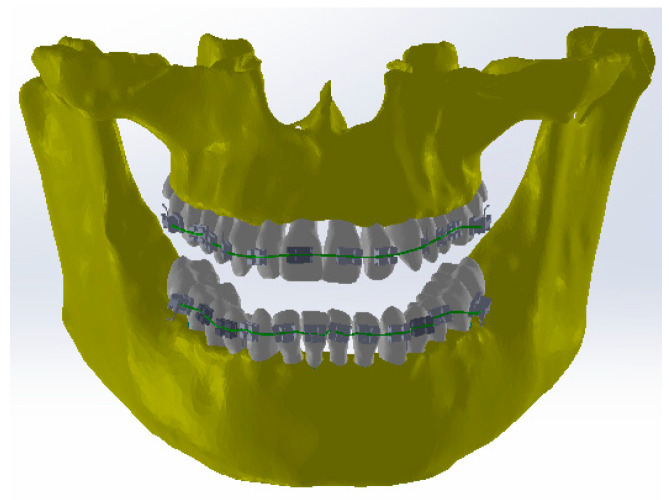
The identical virtual model of the real orthodontic system.

**Figure 18 diagnostics-13-01567-f018:**
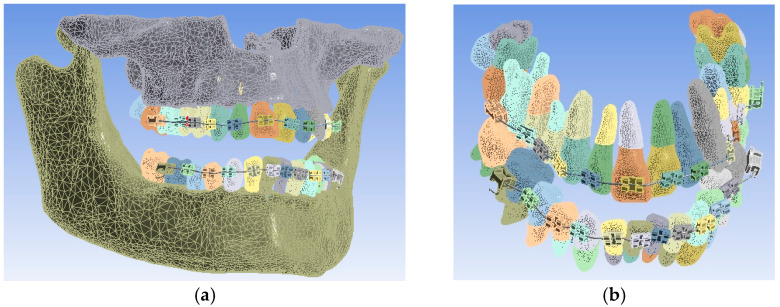
The model of the orthodontic system loaded in Ansys Workbench: complete orthodontic model (**a**); the orthodontic model with the bone components temporarily suppressed (**b**).

**Figure 19 diagnostics-13-01567-f019:**
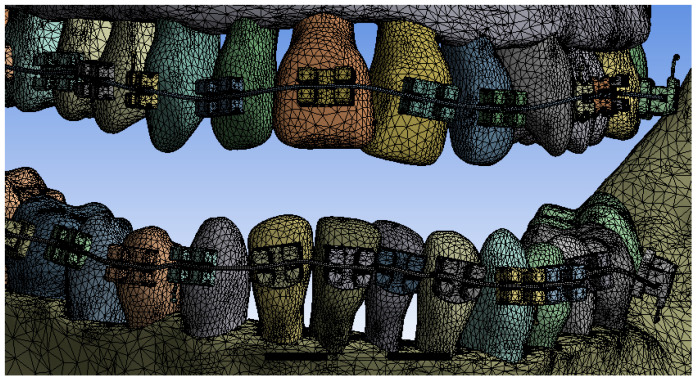
Finite element structure of the studied system.

**Figure 20 diagnostics-13-01567-f020:**
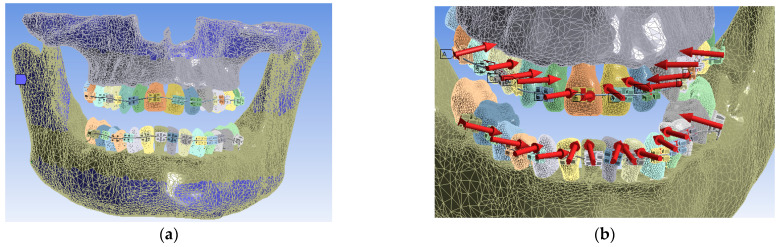
The fixation surfaces (**a**) and the system of forces used in the simulation of the orthodontic assembly (**b**); A to J indicate the order of force definition in the software; the red arrows show the position and orientation of the forces in the direction given by the deformation of the orthodontic wire.

**Figure 21 diagnostics-13-01567-f021:**
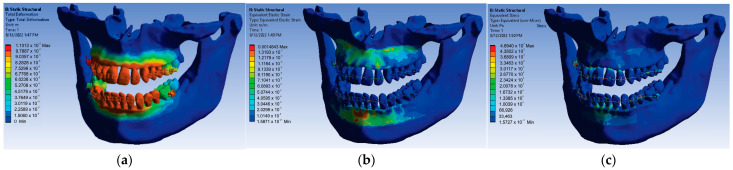
Displacement (**a**); strain (**b**); stress (**c**) maps for 0.5 N.

**Figure 22 diagnostics-13-01567-f022:**
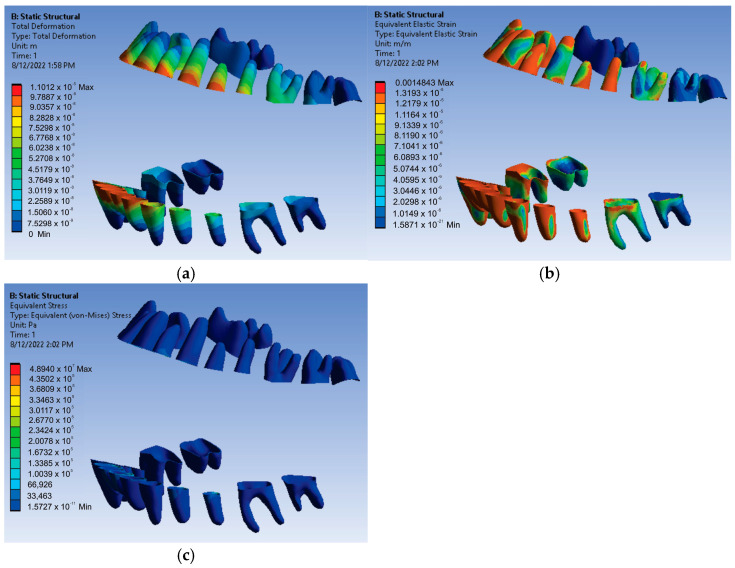
Result maps: displacement (**a**); strain (**b**); and stress (**c**) for 0.5 N.

**Figure 23 diagnostics-13-01567-f023:**
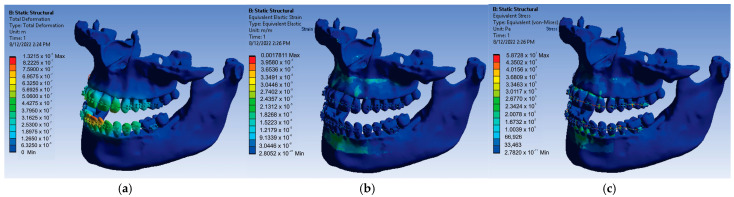
Result maps: displacement (**a**); strain (**b**); and stress (**c**) for 0.6 N.

**Figure 24 diagnostics-13-01567-f024:**
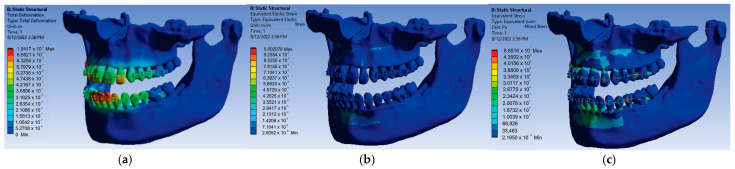
Result maps: displacement (**a**); strain (**b**); and stress (**c**) for 0.7 N.

**Figure 25 diagnostics-13-01567-f025:**
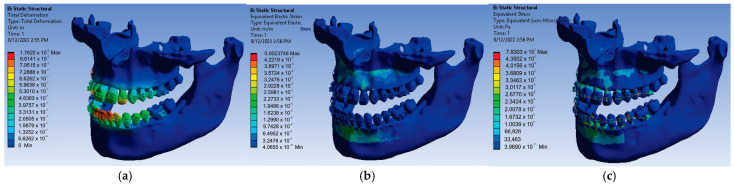
Result maps: displacement (**a**); strain (**b**); and stress (**c**) for 0.8 N.

**Figure 26 diagnostics-13-01567-f026:**
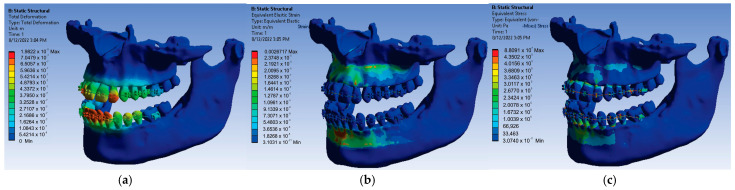
Result maps: displacement (**a**); strain (**b**); and stress (**c**) for 0.9 N.

**Figure 27 diagnostics-13-01567-f027:**
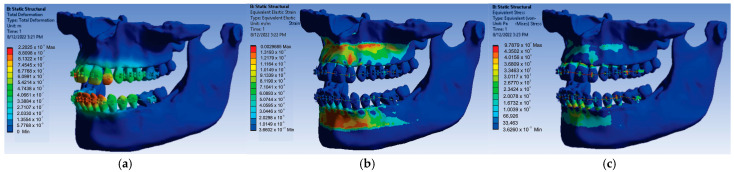
Result maps: displacement (**a**); strain (**b**); and stress (**c**) for 1 N.

**Figure 28 diagnostics-13-01567-f028:**
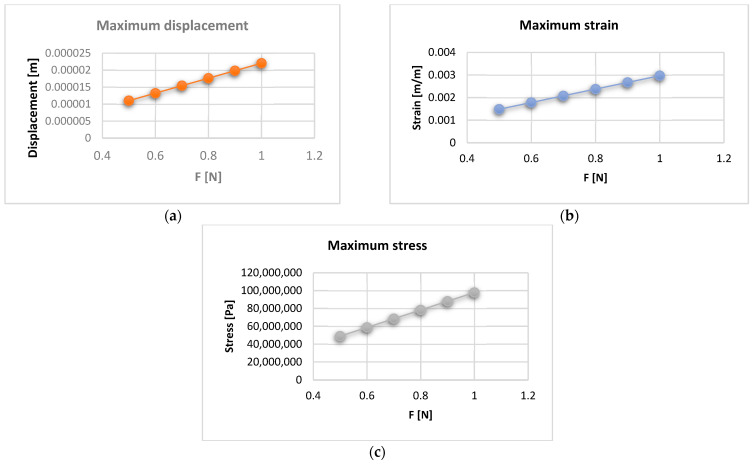
Diagram for the maximum displacement (**a**); strain (**b**); and stress (**c**) values.

**Figure 29 diagnostics-13-01567-f029:**
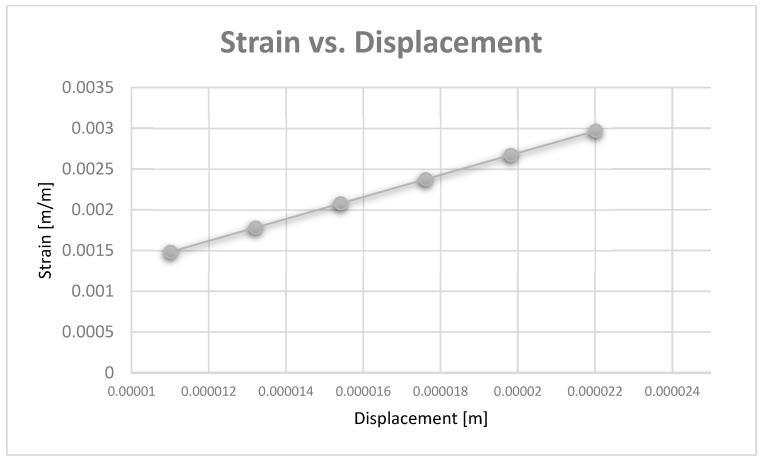
Strain vs. displacement diagram.

**Figure 30 diagnostics-13-01567-f030:**
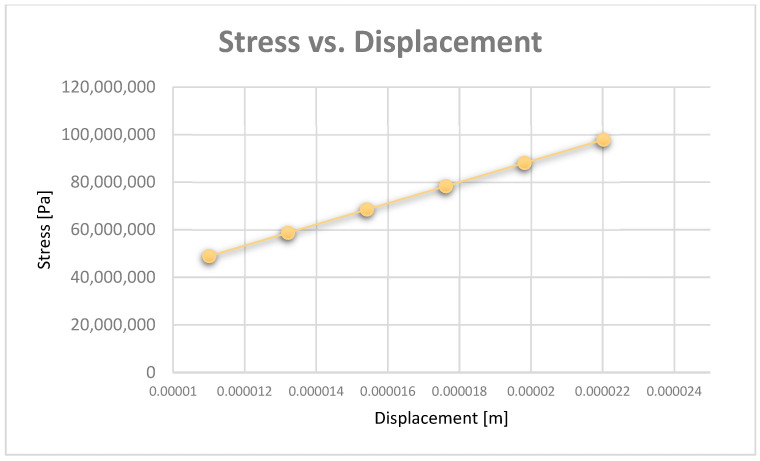
Stress vs. displacement diagram.

**Table 1 diagnostics-13-01567-t001:** Physical properties of materials used in the simulations.

Component	Material	Density	Young’s Modulus	Transverse Modulus of Elasticity	Poisson’s Coefficient
Bracket- and tube-type elements	Ni + Cr alloy	8500 Kg/m^3^	2.1 × 10^11^ Pa	8.015 × 10^10^ Pa	0.31
Orthodontic wires	Nitinol	6450 Kg/m^3^	8.3 × 10^7^ Pa	3.12 × 10^7^ Pa	0.33
Maxillary, mandible	Bone	1400 Kg/m^3^	1 × 10^10^ Pa	3.84 × 10^9^ Pa	0.31
Periodontal ligaments	Ligament	1100 Kg/m^3^	0.62 × 10^9^ Pa	0.23 × 10^9^	0.42
Teeth	Enamel	2958 Kg/m^3^	7.79 × 10^10^ Pa	2.996 × 10^10^ Pa	0.3

## Data Availability

The authors declare that the data from this research are available from the corresponding authors upon reasonable request.

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
