# Peer review of "Using the Finite Element Method to Determine the Odonto-Periodontal Stress for a Patient with Angle Class II Division 1 Malocclusion"

_diagnostics, 2023, doi:10.3390/diagnostics13091567_

Round 1
Reviewer 1 Report
This paper describes Using the Finite Element Method to Determine the Odonto-Periodontal Stress in an orthodontic patient.
I have no major complaints, and the work is well-written.
What I would recommend is that the work be proofread by a native speaker or mdpi professional proofreading because there are still small mistakes in grammar and style,
This paper describes Using the Finite Element Method to Determine the Odonto-Periodontal Stress in an orthodontic patient.
I have no major complaints, and the work is well-written.
What I would recommend is that the work be proofread by a native speaker or mdpi professional proofreading because there are still small mistakes in grammar and style,
Author Response
What I would recommend is that the work be proofread by a native speaker or mdpi professional proofreading because there are still small mistakes in grammar and style.
Thank you for taking the time to evaluate our article for publication! Our article was proofread by a native English speaker.
Reviewer 2 Report
Dear Authors,
thank you for an interesting way of preparation of a case report. Here are some suggestions of mine:
1. The introduction is more like "divided into independent parts", it should be led more "merged", so that it looks like one piece in general. A very interesting point refers to WHO - to my mind it should be widened by 1-2 more sentences.
2. When you write about the use of CBCT to diagnosis, I would add more of that, because this examination shows us without doubt the volumetric of orophatungeal airways and the correlation with the occlussion, see:
- Vidal-Manyari PA, Arriola-Guillén LE, Jimenez-Valdivia LM, Dias-Da Silveira HL, Boessio-Vizzotto M. Effect of the application of software on the volumetric and cross-sectional assessment of the oropharyngeal airway of patients with and without an open bite: A CBCT study. Dent Med Probl. 2022;59(3):397–405. doi:10.17219/dmp/145287
- Rathi S, Gilani R, Kamble R, Bhandwalkar S. Temporomandibular Joint Disorder and Airway in Class II Malocclusion: A Review. Cureus. 2022 Oct 20;14(10):e30515. doi: 10.7759/cureus.30515.
3. Lines 98-99, please provide the camera used for photos (if used a phone camera, please add the explanation in comparison to SLR)
4. Lines 103-105, please add what apparatus was used (company, country of production etc), please add the parameters - no information of CBCT had been added
5. fig 3 should go before fig. 2 (this is how the text of the article is formed), there is no reference in the main text to the images presented.
6. Please, note other mathematical methods for establishing bone levels, eg. fractal dimension analysis
- Jurczyszyn K, Kubasiewicz-Ross P, Nawrot-Hadzik I, Gedrange T, Dominiak M, Hadzik J. Fractal dimension analysis a supplementary mathematical method for bone defect regeneration measurement. Ann Anat. 2018 Sep;219:83-88. doi: 10.1016/j.aanat.2018.06.003.
7. The conclusions are not clear to me - please redraft it.
To sum up, the paper has to be revised
Author Response
Thank you for taking the time to evaluate our article for publication!
- The introduction is more like "divided into independent parts", it should be led more "merged", so that it looks like one piece in general. A very interesting point refers to WHO - to my mind it should be widened by 1-2 more sentences.
We made some improvements in the introduction as you suggested.
We widened the part from the introduction regarding the WHO as you suggested:
“Also, the WHO refers to malocclusions as a disability due to the impairment of the normal functions of the dento-maxillary apparatus, particularly physiognomy. In this way, the stress to which the individual and his family are subjected was solved. In modern man, the correction of malocclusions has become a necessity in his social behavior. Studies in the specialized literature have shown that a patient esthetically and functionally rehabilitated through orthodontic treatment became much more socially attractive than one with malocclusion [3]. A successful orthodontic therapy aims to achieve a correlation between facial aesthetics and normal occlusion.”
- When you write about the use of CBCT to diagnosis, I would add more of that, because this examination shows us without doubt the volumetric of oropharyngeal airways and the correlation with the occlussion, see:
We took into account the references you suggested:
- Vidal-Manyari PA, Arriola-Guillén LE, Jimenez-Valdivia LM, Dias-Da Silveira HL, Boessio-Vizzotto M. Effect of the application of software on the volumetric and cross-sectional assessment of the oropharyngeal airway of patients with and without an open bite: A CBCT study. Dent Med Probl. 2022;59(3):397–405. doi:10.17219/dmp/145287
Another study performed on two groups of 60 cases selected from 137 CBCT scans obtained from patients with and without an open bite (each group including adults of both genders and approximately equal mean age) demonstrated how two different three-dimensional software packages (Planmeca Romexis and Nemotec NemoStudio) used in CBCT interpretation affect the volumetric and cross-sectional measurements of the oropharyngeal airway, particularly in individuals without an open bite [7].
- Rathi S, Gilani R, Kamble R, Bhandwalkar S. Temporomandibular Joint Disorder and Airway in Class II Malocclusion: A Review. Cureus. 2022 Oct 20;14(10):e30515. doi: 10.7759/cureus.30515.
Also, recent research indicates the correlation between skeletal Class II malocclusion and the position of the mandibular condyle affecting the temporomandibular joint (TMJ) and breathing. The authors of this study emphasize the importance of starting treatment of TMJ disorders as soon as this link is noted [8].
- Lines 98-99, please provide the camera used for photos (if used a phone camera, please add the explanation in comparison to SLR)
The photos were made with a DSLR 600EOS camera (Canon, Ota, Tokyo, Japan).
- Lines 103-105, please add what apparatus was used (company, country of production etc), please add the parameters - no information of CBCT had been added.
We added details about the CT used in our study. “To obtain tomographic images, we used a CS 8200 3D CT scanner (Carestream Dental, Atlanta, GA, SUA).”
- fig 3 should go before fig. 2 (this is how the text of the article is formed), there is no reference in the main text to the images presented.
We put fig. 3 before fig. 2 as you suggested. We have reference in the main text to the images presented.
- Please, note other mathematical methods for establishing bone levels, eg. fractal dimension analysis
- Jurczyszyn K, Kubasiewicz-Ross P, Nawrot-Hadzik I, Gedrange T, Dominiak M, Hadzik J. Fractal dimension analysis a supplementary mathematical method for bone defect regeneration measurement. Ann Anat. 2018 Sep;219:83-88. doi: 10.1016/j.aanat.2018.06.003.
There are also mathematical methods for establishing bone levels, such as fractal dimension analysis (FDA). The geometry of bones is very complex. Its shape is complicated to measure using only traditional measuring methods. Traditional measuring techniques of a bone specimen require supplementary measuring. For the FDA, mathematic formulas are used to describe complicated shapes. The FDA offers a possibility of a comparison between complex shapes such as a histological image of a bone defect.
- The conclusions are not clear to me - please redraft it.
We redrafted the conclusions as you suggested:
“Through this study, we demonstrated that functional correlations can be determined between the values of the forces applied on the bracket and tube elements and the maximum values of displacement, strain and stress.
The maximum values from the stress maps, which indicate the areas of possible damage, are relatively small, which proves once again that the treatment based on the straight-wire technique cannot lead to damage to the dental and periodontal tissues, if it is applied according to the known orthodontic protocols.
From the analyzes carried on the stress diagrams, it can be also seen that the maximum values are found on the buccal surfaces of the teeth, just below the bracket and tube elements. This observation indicates that enamel cracks may appear in these areas, if the fixation of bracket and tube elements is not done correctly or if the orthodontic system based on the straight-wire technique is maintained for a longer period than is established in the treatment plan.”
Reviewer 3 Report
The present paper addresses a topic of great interest to both researchers and clinicians. However, minor revisions are needed.
-The "Introduction" section is insufficient. It is necessary to report the state of the art on the topic addressed, and what the literature says about the chosen topic. A detailed analysis of the gaps present in the literature is missing. Please, report the outcomes extracted from the literature in reference to the objectives of your study.
-It is necessary to insert at least one null hypothesis at the end of the introduction section.
-What about the criteria of inclusion of the patient included in the present study?
-The "Discussion" section should begin by stating whether the null hypothesis has been accepted or rejected.
-The "Discussion" section itself is unacceptable as it stands now. There are no sufficient explanations to justify the results obtained. Furthermore, comparisons with the results present in the literature are missing and a paragraph regarding future study perspectives is missing.
-A detailed paragraph regarding the limitations of this clinical study is mandatory in the "Discussion" section.
-The "Conclusions" section cannot be so short. It needs to be enriched and discursive.
-The list of References is scarce and many recent papers are missing. Please update the reference list.
-The quality of images is poor
-There are too many figures
-English revision is required by a native English speaker.
Author Response
Thank you for taking time to evaluate our article for publication!
-The "Introduction" section is insufficient. It is necessary to report the state of the art on the topic addressed, and what the literature says about the chosen topic. A detailed analysis of the gaps present in the literature is missing. Please, report the outcomes extracted from the literature in reference to the objectives of your study.
Studying the specialized literature, no significant research was found to analyze the mechanical behavior of an orthodontic system. However, even if a FEM study is done only on one tooth, it is highlighted that the forces in the orthodontic system have a value of 1 N or less [11]. Also, similar conclusions resulted from another study, the force values that appear being approximately 1 N [12]. At the same time, in a recent research, a system consisting of a simplified mandible and four teeth was analyzed, testing forces between 0.25 and 5 N [14]. These are three studies were carried out on simple three-dimensional models, so not on systems similar to those of the patients, having mainly theoretical significance. A more complex study, carried out on a model similar to the model of a patient, determined the forces that appear in the entire orthodontic system, having a value between 0 and 5 N [15].
-It is necessary to insert at least one null hypothesis at the end of the introduction section.
“Also, we were interested in checking if there is a functional correlation between the value of forces acting the brackets and the maximum values of displacement, strain and stress and if the application of forces through the orthodontic wire can cause damage to the periodontal ligament.”
-What about the criteria of inclusion of the patient included in the present study?
“The inclusion criteria in our study were based on the complexity of the clinical picture of the selected patient, who was diagnosed with Angle Class II Division 1 malocclusion and maxillary compression with protrusion. The tooth movements that occur during orthodontic treatment with fixed appliances are essential in determining the behavior of the entire orthodontic system.”
-The "Discussion" section should begin by stating whether the null hypothesis has been accepted or rejected.
“In the diagrams shown in figures 28-30, we illustrated the the approximately linear correlation between displacement, strain and stress. We also showed the value of the applied forces. At the same time, a functional link was determined between displacement, strain and stress, which shows that the forces that appear in an orthodontic system produce effects that cannot lead to damage of the periodontal ligament.”
-The "Discussion" section itself is unacceptable as it stands now. There are no sufficient explanations to justify the results obtained. Furthermore, comparisons with the results present in the literature are missing and a paragraph regarding future study perspectives is missing.
“Although there are few research that use FEM to study the dentomaxillary apparatus based on orhodontic wires, the values of the forces are similar to the forces used in our study. Also, the effects of of their application is almost the same, rendered by displacement, strain and stress maps. However, our study is based on a much more complicated system, similar to the system of a real patient, where the geometry of the virtual tissues is quasi-identical, being obtained from the processing of real CBCT images. Also, in this system, the periodontal ligament was also included in order to obtain a three-dimensional model as close as possible to reality.”
-A detailed paragraph regarding the limitations of this clinical study is mandatory in the "Discussion" section.
“The limitations of our study are represented by the fact that not all the soft tissues were taken into account (such as dental pulp and gingival tissue), because, for now, we considered that their effect is not significant.”
-The "Conclusions" section cannot be so short. It needs to be enriched and discursive.
We enriched the conclusions as you suggested:
“Through this study, we demonstrated that functional correlations can be determined between the values of the forces applied on the bracket and tube elements and the maximum values of displacement, strain and stress.
The maximum values from the stress maps, which indicate the areas of possible damage, are relatively small, which proves once again that the treatment based on the straight-wire technique cannot lead to damage to the dental and periodontal tissues, if it is applied according to the known orthodontic protocols.
From the analyzes carried on the stress diagrams, it can be also seen that the maximum values are found on the buccal surfaces of the teeth, just below the bracket and tube elements. This observation indicates that enamel cracks may appear in these areas, if the fixation of bracket and tube elements is not done correctly or if the orthodontic system based on the straight-wire technique is maintained for a longer period than is established in the treatment plan.”
-The list of References is scarce and many recent papers are missing. Please update the reference list.
We updated the references with some recent papers.
-The quality of images is poor
We improved the quality of the images.
-There are too many figures
All the figures used are relevant for our study.
-English revision is required by a native English speaker.
Our article was proofread by a native English speaker.
Round 2
Reviewer 2 Report
The changes had been applied. Thank you